# Heterogeneity in response to serological exposure markers of recent *Plasmodium vivax* infections in contrasting epidemiological contexts

Jason Rosado[1,2], Michael T. White[1], Rhea J. Longley[3,4], Marcus Lacerda[5,6], Wuelton Monteiro[6], Jessica Brewster[3], Jetsumon Sattabongkot[7], Mitchel Guzman-Guzman[8], Alejandro Llanos-Cuentas[9], Joseph M. Vinetz[8,9,10,11], Dionicia Gamboa[8,11], Ivo Mueller[1,3,4]*

1 Unit of Malaria: Parasites and hosts, Institut Pasteur, Paris, France, 2 Sorbonne Université, ED 393, Paris, France, 3 Walter and Eliza Hall Institute of Medical Research, Melbourne, Australia, 4 Department of Medical Biology, University of Melbourne, Australia, 5 Instituto Leônidas & Maria Deane (Fiocruz), Manaus, Brazil, 6 Tropical Medicine Foundation Dr Heitor Vieira Dourado, Manaus, Amazonas, Brazil, 7 Vivax Research Unit, Faculty of Tropical Medicine, Mahidol University, Bangkok, Thailand, 8 Laboratorio ICEMR-Amazonia, Laboratorios de Investigación y Desarrollo, Facultad de Ciencias y Filosofía, Universidad Peruana Cayetano Heredia, Lima, Peru, 9 Instituto de Medicina Tropical Alexander von Humboldt, Universidad Peruana Cayetano Heredia, Lima, Peru, 10 Section of Infectious Diseases, Department of Internal Medicine, Yale School of Medicine, New Haven, Connecticut, United States of America, 11 Departamento de Ciencias Celulares y Moleculares, Facultad de Ciencias y Filosofía, Universidad Peruana Cayetano Heredia, Lima, Peru

* mueller@wehi.edu.au

**Data Availability Statement:** All relevant data are within the manuscript and its Supporting Information files.

## Abstract

### Background

Antibody responses as serological markers of *Plasmodium vivax* infection have been shown to correlate with exposure, but little is known about the other factors that affect antibody responses in naturally infected people from endemic settings. To address this question, we studied IgG responses to novel serological exposure markers (SEMs) of *P. vivax* in three settings with different transmission intensity.

### Methodology

We validated a panel of 34 SEMs in a Peruvian cohort with up to three years' longitudinal follow-up using a multiplex platform and compared results to data from cohorts in Thailand and Brazil. Linear regression models were used to characterize the association between antibody responses and age, the number of detected blood-stage infections during follow-up, and time since previous infection. Receiver Operating Characteristic (ROC) analysis was used to test the performance of SEMs to identify *P. vivax* infections in the previous 9 months.

### Principal findings

Antibody titers were associated with age, the number of blood-stage infections, and time since previous *P. vivax* infection in all three study sites. The association between antibody

**Funding:** This work was supported by National Health and Medical Research Council Australia grants 1092789, 1134989 and 1043345 to IM (https://www.nhmrc.gov.au/), the Global Health Innovative Technology Fund grant T2015-142 to IM (https://www.ghitfund.org/), National Institutes of Health-National Institute of Allergy and Infectious Diseases (NIH-NIAID) U19AI089681 to JMV (https://www.niaid.nih.gov). This work has been supported by FIND with funding from the Australian and British governments. The Brazilian team was partly funded by Fundação de Amparo à Pesquisa do Estado do Amazonas-FAPEAM (PAPAC 005/2019 and Pró-Estado). JR is supported by the Pasteur - Paris University (PPU) International PhD Program. RJL received the Page Betheras Award from WEHI to provide funding for technical support for this project during parental leave. RJL is supported by a NHMRC Early Career Investigator Fellowship (1173210). ML and WM are fellows of the Brazilian National Council for Scientific and Technological Development. MGG is supported by Training Grant 5D43TW007120 (https://www.fic.nih.gov). The funders had no role in study design, data collection and analysis, decision to publish, or preparation of the manuscript.

**Competing interests:** I have read the journal's policy and the authors of this manuscript have the following competing interests: RJL, MTW, Takafumi Tsuboi and IM are inventors on patent PCT/US17/67926 on a system, method, apparatus and diagnostic test for Plasmodium vivax. No other authors declare a conflict of interest.

titers and time since previous *P. vivax* infection was stronger in the low transmission settings of Thailand and Brazil compared to the higher transmission setting in Peru. Of the SEMs tested, antibody responses to RBP2b had the highest performance for classifying recent exposure in all sites, with area under the ROC curve (AUC) = 0.83 in Thailand, AUC = 0.79 in Brazil, and AUC = 0.68 in Peru.

## Conclusions

In low transmission settings, *P. vivax* SEMs can accurately identify individuals with recent blood-stage infections. In higher transmission settings, the accuracy of this approach diminishes substantially. We recommend using *P. vivax* SEMs in low transmission settings pursuing malaria elimination, but they are likely to be less effective in high transmission settings focused on malaria control.

## Author summary

*Plasmodium vivax* still poses a threat in many countries due to its ability to cause recurrent infections. Key to achieving the goal of malaria elimination is the ability to quickly detect and treat carriers of relapsing parasites. Failing to identify this transmission reservoir will hinder progress towards malaria elimination. Recently, novel serological markers of recent exposure to *P. vivax* (SEM) have been developed and validated in low transmission settings. It is still poorly understood what factors affect the antibody response to these markers when evaluated in contrasting endemic contexts. To determine the factors that influence the antibody response to SEM, we compared the antibody levels in three sites with different transmission intensity: Thailand (low), Brazil (moderate) and Peru (high). In this study, we found that transmission intensity plays a key role in the acquisition of the antibody repertoire to *P. vivax*. In highly endemic sites, it is likely that immunological memory resulting from a constant and sustained exposure will impact the performance of SEMs to detect individuals with recent exposure to *P. vivax*. In summary, SEMs that perform well in low transmission sites do not perform as well in high transmission regions.

## Introduction

*Plasmodium vivax* is the most geographically widespread *Plasmodium* species and the second largest cause of clinical malaria worldwide. Although the global burden has decreased from an estimated 24.5 million cases in 2000 to 7.5 million cases in 2018, *P. vivax* remains a challenging parasite to control and eliminate due to its biology, notably its ability to relapse from dormant liver-stage hypnozoites [1], and its high transmission potential caused by rapid production of gametocytes and short development time in the mosquito vector [2].

In tropical regions, most *P. vivax* relapses occur within 9 months of the initial mosquito bite, with longer intervals observed in temperate regions and parts of the sub-tropics [1]. Except in India, relapses typically account for greater than 80% of all *P. vivax* blood-stage infections [3]. Moreover, in cross-sectional surveys 90–100% of infections are asymptomatic [4] and up to 67% of infections are not detected by conventional screening tools such as light microscopy [5] because of low parasite density [6,7]. Although the role of asymptomatic infections is not well understood, all *P. vivax* blood-stage infections produce gametocytes and may contribute to maintaining transmission [8].

As the number of malaria cases in a region decreases and transmission becomes more heterogeneous, monitoring and mapping of residual transmission pockets becomes increasingly important. Many countries have declared the ambitious goal of eliminating malaria by 2030, and thus, a tool able to identify residual transmission or document the absence of recent transmission is urgently needed. This tool would ideally detect the entire *P. vivax* infectious reservoir composed of individuals with asymptomatic blood-stage infections as well as silent hypnozoite carriers.

Light microscopy (LM) and qPCR, although informative, are imperfect tools due to the poor sensitivity for detecting low density blood-stage infections LM and the inability to detect hypnozoite carriers (LM and qPCR). The antibody responses mounted to *P. vivax* can last weeks to several years after exposure [9,10], making them an ideal tool for assessing transmission history. Thus, by exploiting antibody longevity to assess recent exposure, we can use antibody levels to estimate the time since previous exposure and potentially detect hypnozoite carriers [11].

Serology has been used to estimate malaria risk and endemicity at the population level, detecting temporal and spatial variation in malaria exposure [12], and evaluating malaria control efforts in areas where transmission has decreased to low levels [13]. Therefore, serological markers are an appealing tool in the context of malaria elimination because population-level serological signatures can provide insights of both current and recent exposure to malaria when parasite prevalence surveys are no longer (cost-)efficient because of the small proportion of individuals with detectable blood-stage infection.

The lack of standardized methods and antigens has hindered the implementation of serology as a tool for malaria surveillance. Recently, novel Serological Exposure Markers (SEM) to *P. vivax* have been developed and validated in low transmission settings showing a promising application in detecting likely hypnozoite carriers [11]. Their performance at higher transmission levels is yet to be clarified. It is particularly important to understand what factors cause variation in SEM's performance before their application at a population level. Here, using a sero-epidemiological approach, we compare the performance of SEMs to detect recent exposure to *P. vivax* in three different transmission settings. Epidemiological factors such as age, number of detected blood-stage infections and time since the previous infection were evaluated to understand how they affect SEM performance.

## Methods

### Ethics

The Peruvian cohort was approved by the Institutional Ethics Committee from the Universidad Peruana Cayetano Heredia (UPCH) (SIDISI 57395/2013) and from the University of California San Diego Human Subjects Protection Program (Project # 100765). UPCH also approved the use of the Peruvian serum samples in the Walter and Eliza Hall Institute of Medical Research (WEHI) (SIDISI 100873/2017). The Thai cohort was approved by the Ethics Committee of the Faculty of Tropical Medicine, Mahidol University, Thailand (MUTM 2013-027-01). The Brazilian study was approved by the FMT-HVD (51536/2012) and by the Brazilian National Committee of Ethics (CONEP) (349.211/2013). The Human Research Ethics Committee (HREC) at WEHI approved samples for use in Melbourne (#14/02).

### Field studies

The Peruvian cohort was conducted in two Amazonian villages in the Loreto Region: San José de Lupuna, and Cahuide [14]. Lupuna is located 10 km from Iquitos district (03˚44.591′S, 73˚19.615′W), a forested area only accessible by river. Cahuide (04˚13.785′ S, 73˚276′ W) is

located 60 km from Iquitos city on the Iquitos-Nauta road. Villagers work mainly in agriculture, fishing and occasional hunting. In 2016, malaria cases in Peru represented 14.3% of all cases in South America, from which 96% were reported from Loreto Region [15,16]. Malaria cases have been steadily increasing since 2010–2011, after the cessation of the international financial support program "PAMAFRO", and worsening due to the Loreto flood between 2012–2013 that inundated and damaged many riverine communities [17]. Transmission is stable in both Lupuna and Cahuide, with a peak season from November to May [14].

A three year-long observational cohort study was conducted over December 2012 –December 2015 in Loreto, Peru. Using home-to-home and community-based screening, volunteers $\geq$ 3 years old were invited to participate in this cohort. Participants excluded from the cohort had the following characteristics: being younger than three years old and not giving informed consent to use a blood sample. In July–August 2012, a census was administered to 1007 and 1440 people in Lupuna and Cahuide, respectively. In the enrollment visit, 2197 people were scheduled for follow up from January 2013 to December 2015. The sample size estimation assumed the following: 20% residents had at least 1 microscopically confirmed malaria infection annually; 2% precision; 25% loss to follow-up; 80% power; and 5% significance level [18]. This cohort initially enrolled 1029 participants that were all followed up with monthly population (community) screening (mPS), weekly active case detection (wACD) and continuous passive case detection (PCD) for 12 months. A subsample of 456 individuals was followed up with monthly mPS for another 24 months. Passive case detection relied on care-seeking behavior of individuals with malaria-compatible symptoms at health posts, where axillary temperature and microscopy-directed treatment was done. The mPS were conducted in the first week of each calendar month and involved finger prick-blood sampling for microscopy and dried blood spots. The home-to-home screenings relied on wACDs in the remaining weeks of each month and collected blood samples for microscopy from participants who had malaria-compatible symptoms within the past 7 days. Only participants with microscopically confirmed infections during PCD or ACD were treated with Chloroquine for 3 days (10 mg/g on days 1 and 2, and 5 mg/kg on day 3) plus primaquine for 7 days (0.5 mg/kg per day). Treatment adherence was monitored by trained staff in health posts when in PCD. In contrast to PCD patients, ACD patients self-reported a full treatment adherence [18]. In our study, we selected those participants with serum sample collected at the end of year 3, and at least complete qPCR data for the last 12 months of follow-up (n = 590). This subset included 456 participants with at least 35 qPCR results in the three-year follow-up and 244 participants with at least 12 qPCRs results in the last year of follow-up (37 active case detection visits in total) (S1 Fig). The participants with at least 12 qPCRs results were participants with incomplete blood samples in the first two years of follow-up. The PCR prevalence at the beginning of the cohort was 16% for *P. vivax* and 2% for *P. falciparum* [19].

Data from the Thai and Brazilian cohorts had been reported previously [11]. Briefly, year-long observational cohort studies were conducted over 2013–2014 in Kanchanaburi/Ratchaburi provinces, Thailand [20], and in Brasileirinho, Ipiranga and Puraquequara, three peri-urban communities in Manaus province, Amazonas State [21]. 999 volunteers of all ages were enrolled from Thailand and sampled every month over the year-long cohort, with 14 active case detection visits performed in total. 826 volunteers attended the final visit (S1 Fig). From 2400 inhabitants surveyed in November 2012 in the three communities of Manaus, a total of 1274 residents of all ages were enrolled in the study in April 2013, and sampled every month over the year-long period, with 13 active case detection visits performed in total. 928 volunteers attended the final visit with plasma from 925 available (S1 Fig). At the end of the follow-up, the PCR prevalence of *P. vivax* was 3.0% in Thailand, and 4.2% in Brazil (S1 Table), while the cumulative PCR prevalence of *P. vivax* was 11.7% and 25.43% in Thailand and Brazil,

respectively. All cohort participants provided written informed consent for both participation in the study and future use of samples in studies of antimalarial antibody responses. Parental written consent and written assent were obtained in case of child participants.

### Sample collection and molecular diagnosis

In the Peruvian cohort, blood samples were collected by finger prick onto filter paper and left to dry at room temperature. Dried blood samples were stored at -20˚C prior to molecular diagnosis. In the last visit, whole blood samples were collected and serum was separated by centrifugation and kept at -80˚C until processing. DNA was isolated from dried blood samples using the E.Z.N.A. Blood DNA Mini Kit (Omega Bio-tek, Inc., Norcross, GA, US), according to the manufacturer's instructions. Subsequent amplification was performed by a real-time quantitative PCR (qPCR) method using PerfeCTa SYBR Green FastMix 1250 (Quanta bio, Beverly, MA, US) as previously described [19]. In the Thai and Brazil cohorts, malaria parasites were detected by qPCR as previously described [8,22].

### Multiplex serological evaluation

Serum samples from the Peruvian cohort were evaluated following the protocol reported by Longley *et al* [23]. Briefly, proteins were coupled to non-magnetic microspheres. Protein-coupled microspheres were incubated with plasma (1/100 dilution in phosphate-buffered saline containing 1% bovine serum albumin and 0.05% (v/v) Tween-20, denoted as PBT) for 30 min at room temperature. After washing, microspheres were incubated for 15 minutes at room temperature with an R-Phycoerythrin (R-PE) -conjugated Donkey Anti-Human IgG antibody (cat#709-116-098; JacksonImmunoResearch, UK) to detect total IgG (1/100 dilution). Final washing was followed by resuspension of microspheres in 100 μL of PBT. This protocol was previously validated in other cohorts as well as in this cohort by testing a small group of samples in duplicate [11,23,24]. Therefore, serum samples were evaluated in singlicate format (one well per sample).

34 expressed full-length proteins were tested (See S2 Table for details). They were downselected from a larger panel as previously reported [11]. 30 of the expressed proteins corresponded to erythrocytic stages, one to pre-erythrocytic stages, one to sexual stages, and one to a putative protein.

Plasma samples from healthy donors from non-malaria endemic regions (Melbourne, Australia, and Bangkok, Thailand) were used as negative controls (n = 274) [11] (noting that the control set from Rio, Brazil, was not included in the current study). A standard curve made of pooled plasma from hyper-immune Papua New Guinea adults (serial dilutions ranging from 1:50 to 1:51200) was used in each run for quality control and normalization purposes. The antibody measurements were performed in a Milliplex platform based on Luminex technology. Antibody levels were converted from median fluorescence intensity to relative antibody unit (RAU) using a 5-parameter logistic model written in R. Published serological data from the Thai and Brazilian cohorts were compared to Peruvian antibody profiles [11].

### Statistical analysis

All epidemiological data was obtained from the questionnaires or laboratory tests recorded in the databases from the cohorts in Peru, Thailand and Brazil. This study generated new monthly qPCR data and serological data from the final time point from the Peruvian cohort; however, we only used qPCR data of the 13 months preceding the last time-point. We characterized the association between antibody responses and PCR positive blood-stage infections in the Peruvian cohort by fitting multivariate linear regression models adjusted for confounders.

The number of qPCR positive results or blood-stage detections was counted in both the previous six months before end of follow up, and in the first six months of follow up. A Bayesian Information Criterion (BIC) was used to select the model with the best explanatory variables. A retrospective analysis was performed taking into account the last time-point of follow up as a point of reference.

To test the performance of the SEMs to identify recent *P. vivax* exposure, individuals in the cohorts were categorized according to their history of blood-stage infection. Individuals with at least one qPCR positive *P. vivax* sample in the previous 9 months were classified as "recently exposed", whereas those with old infections (> 9 months) or no infections in the time of follow up (13 months for Peru and Brazil, 14 months for Thailand), were classified as "not recently exposed". Nine months was used because this cut-off takes into account that all *P. vivax* strain (with the possible exception of the hybernans strain now restricted to Korea) will experience their first relapse within no more than 9 months following the primary mosquito-bite derived infection [1]. Thus, a 9 months recent exposure period allows to capture almost all people carrying hyponzoites and thus at risk of experiencing a relapse. Receiver operating characteristic (ROC) curves were used to assess the trade-off between sensitivity and specificity of using single SEMs to identify recent infections. Sensitivity was defined as the proportion of people with recent infections with antibody titers higher than a given cut-off value. Specificity was defined as the proportion of subjects without recent infections with antibody titers lowers than the same cut-off value. We used the area under the ROC curve (AUC) value for comparing the test performance given by an antibody response between the three study sites.

Linear discriminant analysis (LDA) was performed to identify the best combinations of antibody responses for classifying recent infections. The algorithm for searching and validating the best antigen combination was described elsewhere [11]. The AUC was calculated for each combination.

To determine the factors which drive variation in antibody responses, multivariate linear regression models were fitted using: (i) log10 (age) (a proxy of lifetime exposure); (ii) number of detected blood-stage infections during 13 months (Peru, Brazil) or 14 months (Thailand) of follow up; (iii) time since previous blood-stage infection; and (iv) an error term accounting for additional unexplained variation. Data on individuals with at least one blood-stage detection during 13 months was utilized for this purpose. The total variance of each marker was normalized to 1 and was the sum of variance given by both explained and unexplained variance, respectively. The contribution of each factor to the total variance was estimated using the "relaimpo" (https://CRAN.R-project.org/package=relaimpo) and "car" (https://CRAN.R-project.org/package=car) R packages [25,26].

Statistical analysis was done using R 3.43 (https://www.r-project.org/) and the R packages MASS, ROCR, rpart, randomForest, and Stata version 12 (StataCorp. 2011. Stata Statistical Software: Release 12. College Station, TX: StataCorp LP.)

## Results

### Epidemiological characteristics of the Peruvian longitudinal cohort

Epidemiological characteristics of the Peruvian study population are summarized in Table 1. A total of 590 individuals had a serum sample at the end of the cohort. Of these individuals, 289 lived in Cahuide (CAH, 48.9%) and 301 lived in Lupuna (LUP, 51.1%). The average age of participants was 29.7 years (range: 3.8–85 years old). Individuals from Cahuide were younger than those from Lupuna (p < 0.01). The overall female/male ratio was 1.39, with similar proportions in both sites (CAH: 1.44, LUP: 1.35). The short duration of residency in the community had previously been identified as a risk factor for *P. vivax* infection in these communities

**Table 1. Epidemiologic characteristics of the newly reported Peruvian study sites and participants.**

| Characteristics | Cahuide (n = 289) | Lupuna (n = 301) | Total (n = 590) |
|---|---|---|---|
| Age (median, range)** | 21.9 (3.80–81.88) | 29.1 (4.44–85.57) | 26.0 (3.80–85.57) |
| <15 years old (%) | 40.48 | 31.56 | 35.93 |
| 15–39 years old (%) | 33.56 | 31.56 | 32.54 |
| ≥40 years old (%) | 25.95 | 36.88 | 31.53 |
| Female sex (%) | 59.16 | 57.47 | 58.30 |
| Time in community, (%) **** | | | |
| <5 years | 26.29 | 3.32 | 14.58 |
| ≥5 years | 73.7 | 96.68 | 85.42 |
| Forest-related occupation (>18 years old) (%) | | | |
| Fishing | 0.62 | 2.67 | 1.73 |
| Woodcutter | 6.83 | 4.81 | 5.76 |
| Farmer* | 25.46 | 35.29 | 30.83 |
| Education (%) | | | |
| None* | 21.45 | 14.95 | 18.13 |
| Primary school | 57.10 | 52.16 | 54.58 |
| Secondary school** | 21.45 | 32.89 | 27.29 |
| Infection by LM at the last time-point | | | |
| *P. vivax* (%) | 2.08 | 1.00 | 1.52 |
| *P. falciparum* (%) | 0 | 0 | 0.00 |
| Infection by qPCR at the last time-point [a] | | | |
| *P. vivax* (%) | 18.68 | 23.58 | 21.19 |
| *P. falciparum* (%) | 2.77 | 2.66 | 2.71 |
| Months of follow-up, *n* (%) | | | |
| 13 months**** | 164 (56.74) | 76 (25.24) | 240 (40.68) |
| 37 months**** | 125 (43.25) | 225 (74.75) | 350 (59.32) |
| *P. vivax* density by qPCR at the last time-point (median, IQR) | 1.60 (0.92–21.61) | 1.39 (0.65–4.96) | 1.55 (0.74–8.38) |
| Total clinical *P. vivax* infections, *n* (%) | | | |
| 0 | 279 (96.54) | 283 (94.01) | 562 (95.25) |
| 1 | 10 (3.46) | 16 (5.31) | 26 (4.41) |
| ≥2 | 0 (0) | 2 (0.66) | 2 (0.34) |
| Participants infected in the last 9 months, *n* (%) **** | 162 (56.05) | 226 (75.08) | 388 (65.76) |
| Total *P. vivax* infections by qPCR in 13 months, *n* (%) | | | |
| 0**** | 98 (33.91) | 48 (15.94) | 146 (24.74) |
| 1**** | 102 (35.29) | 68 (22.59) | 170 (28.81) |
| 2 | 55 (19.03) | 62 (20.60) | 117 (19.83) |
| 3* | 23 (7.96) | 42 (13.95) | 65 (11.02) |
| ≥4**** | 11 (3.80) | 81 (26.91) | 92 (15.59) |
| Time since previous *P. vivax* infection, *n* (%) | | | |
| < 1 month* | 65 (22.49) | 95 (31.56) | 160 (27.12) |
| 1–9 months* | 97 (33.56) | 131 (43.52) | 228 (38.64) |
| 9–13 months | 29 (10.03) | 27 (8.97) | 56 (9.49) |
| No infection in 13 months**** | 98 (33.91) | 48 (15.94) | 146 (24.74) |

[a] Prevalence includes single–and mixed-species infections The last time-point = day of blood and serum sampling for qPCR diagnosis and Luminex. Statistical differences between Cahuide and Lupuna

* p<0.05

** p<0.01

*** p<0.001

**** p<0.0001.

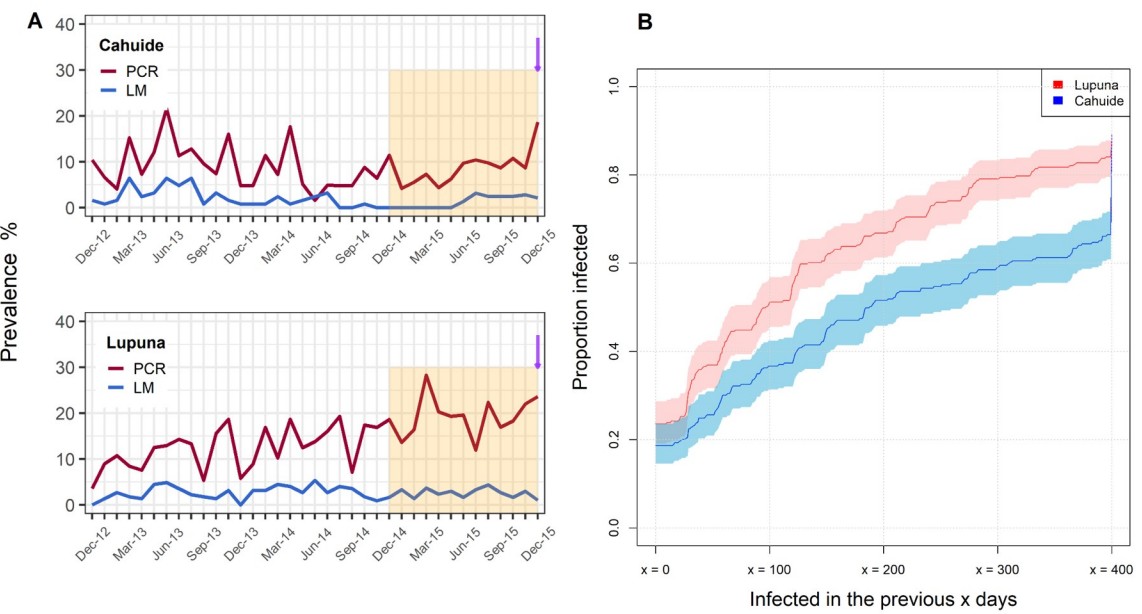

**Fig 1. Prevalence of P. vivax infection in the Peruvian cohort.** (A) Longitudinal prevalence of P. vivax infection by light microscopy or PCR in Cahuide and Lupuna. Retrospective analyses were performed using infection history over the last year of follow-up, which is indicated by the orange shadow. The purple arrow indicates when antibodies were measured. (B) Proportion of individuals with PCR-detectable infection in the previous "x" days the end of follow up. Shaded areas represent the respective 95% confidence intervals.

[19]. Almost 15% (n = 89) of participants had recently settled in the communities (<2 years), with higher proportions in Cahuide (26%) than Lupuna (3%) (p < 0.0001, $\chi^2$ test).

At the last day of follow up, the prevalence of *P. vivax* (Pv) infections detected by qPCR was 21.2% (LUP: 23.6%, CAH: 18.7%) (Fig 1A), of which only 7% were detected by LM (9 patent infections). Age has been described as a risk factor for *P. vivax* exposure at the baseline of this cohort [14]; therefore, we compared the Pv prevalence by qPCR among three age groups at the last time point of follow up. There were no significant differences in prevalence among three defined age groups (p > 0.05). The median parasite density by qPCR was 1.55 parasites/μL (IQR: 0.74–8.38). In the preceding year, December 2014 to December 2015, a total of 7,612 blood samples were collected during follow-up. Of the collected blood samples, 14.2% (1083/7612) were positive for *P. vivax* by qPCR. Of these qPCR positive samples 11.8% (128/1083) were positive by microscopy, and 2.8% (30/1083) exhibited symptoms. In the last 9 months of follow up (the last year of the cohort), 65.6% of individuals experienced at least one blood-stage *P. vivax* infection. There was a significant difference in cumulative PCR prevalence between Cahuide (66.1%) and Lupuna (84.1%) (p < 0.0001, $\chi^2$ test) (Fig 1B).

### Profiles of anti-*P. vivax* antibody responses in Peru

In Lupuna, the geometric mean antibody response to 34 *P. vivax* proteins was 1.93 times higher than in Cahuide, although the distribution of antibody responses overlapped between both communities (0.0014 vs. 0.0007; p < 0.001, t-test, S2 Fig). Henceforth, data from both communities were combined for subsequent analysis. Antibody responses to 26 out of 34 antigens were positively associated with concurrent *P. vivax* infections (Odds ratio range: 1.38–1.99, p < 0.01, S3 Table). Most of the responses to the tested markers showed a positive correlation with the individual's age (spearman correlation factor range ρ = 0.143–0.482; p <0.001,

# Age

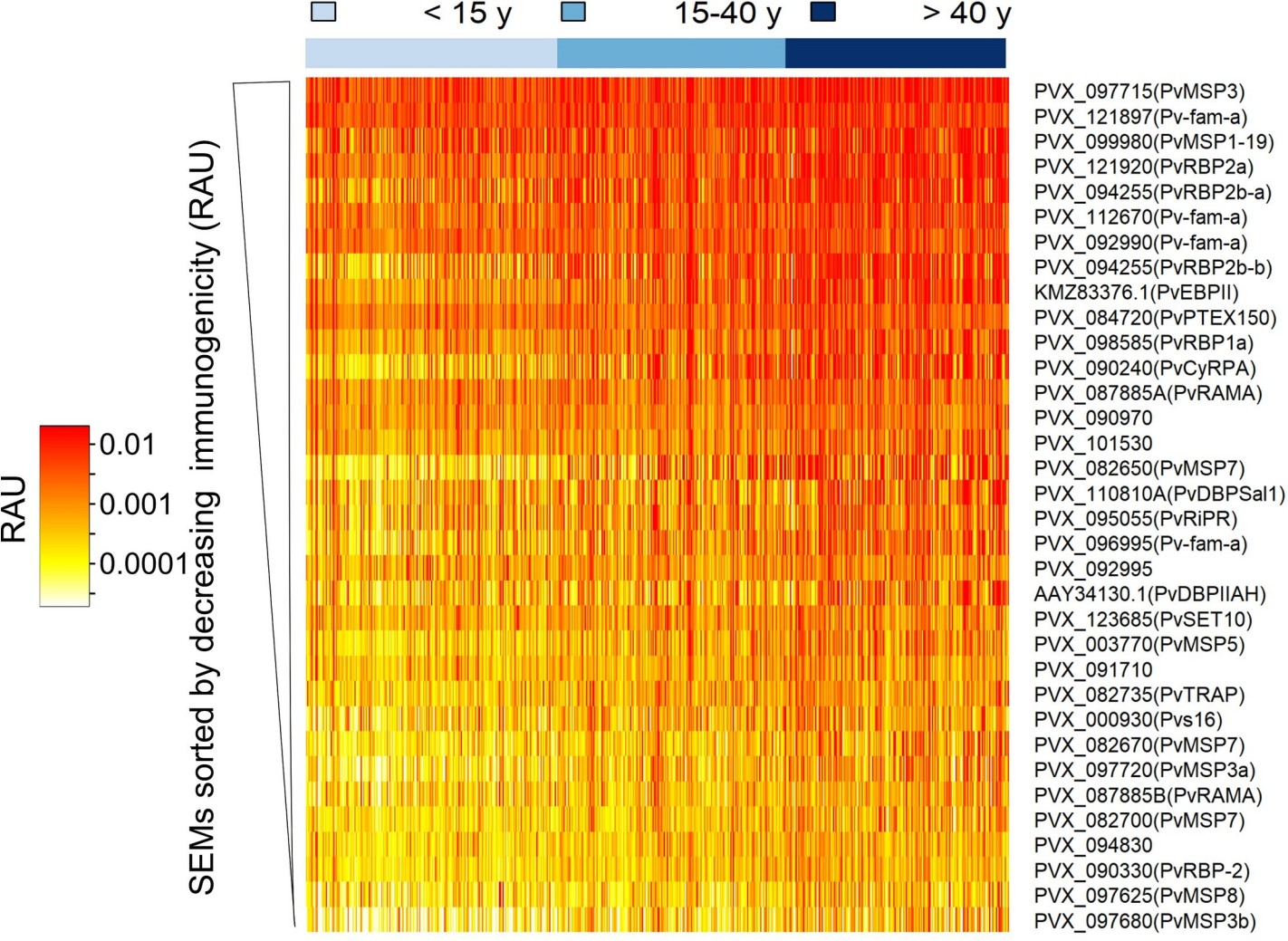

**Fig 2. Heatmap of antibody responses to P. vivax serological exposure markers in the Peruvian cohort.** Relative antibody units (RAU) were log10 transformed. Each vertical line represents a participant sorted by increasing age (3–85 years old). Immunogenicity of SEMs was defined as the geometric mean of log10 (RAU) across all participants in the Peruvian cohort. We used this quantitative metric to sort the antibody responses.

Fig 2, S4 Table), indicating a broader antibody repertoire and higher magnitude of the responses in individuals older than 40 years old.

Antibody responses were strongly correlated where antigens were located in the same parasite location, showing a similar degree of immunogenicity and pattern of reactivity (S3 Fig). For instance, antibody responses to invasion-related proteins (microneme proteins: PVX_110810A (PvDBP region II Sal1 strain), AAY34130.1 (PvDBP region II AH strain), PVX_095055 (PvRiPR), PVX_090240 (PvCyRPA), and KMZ83376.1 (PvEBP region II); and rhoptry proteins: PVX_094255 (PvRBP2b), and PVX_098585 (PvRBP1a)) showed 50 to 98%

correlation to each other, whereas, lower correlations were found with antigens of surface proteins (29–66%).

To identify epidemiological factors that influence the response to SEMs, linear regression models were fitted using demographic and epidemiological data (S5 Table). Age was the strongest explanatory factor for 32 out of 34 proteins (coefficient range: 0.146–1.127, p < 0.01). For instance, a doubling in age will give a 2.18 ($2^{1.127}$) increase in antibody levels to PVX_082650 (PvMSP7). Living in Lupuna and being male were associated with high antibody titers to 20 SEMs (coefficient range: 0.096–0.538, p < 0.01) and 15 SEMs (coefficient range: 0.105–0.202, p < 0.01), respectively, indicating the high exposure in this community. The number of qPCR infections detected in the previous 6 months was an important predictor for 28 antibody responses (coefficient range: 0.042–0.194/infection, p < 0.01), indicating that the level of antibody response to these antigens depends upon the intensity of recent exposure. The antibody responses to PVX_099980 (PvMSP1$_{19}$), PVX_096995 (Pv-fam-a), PVX_094255 (PvRBP2b$_{161-1454}$ and PvRBP2b$_{1986-2653}$), PVX_095055 (PvRiPR) and PVX_087885 were positively associated with the incidence of clinical episodes during the previous 6 months of follow up (coefficient: 0.384–0.505, p < 0.01).

## Transmission intensity and average antibody titers to SEM were higher in Peru than Brazil and Thailand

At the final time point of follow-up, Peru had the highest prevalence of *P. vivax* by qPCR (21.2%, p< 0.001), followed by Brazil (4.2%) and Thailand (3.0%) (S1 Table). During the preceding year of follow up, 75.3% of Peruvian individuals experienced at least one infection of *P. vivax*, followed by Brazil and Thailand where 25.4% and 11.7% of participants had at least one infection. The average antibody titer to all 34 SEMs combined in the Peruvian cohort was 1.64 times higher than Brazil (p < 0.001, t-test) and 2.43 times higher than Thailand (p < 0.001, t-test), confirming the existing high exposure burden in the communities from the Peruvian amazon (See more details in S6 Table). The higher antibody titers in the Peruvian children (< 15 years old) compared with the same age group from Brazil (1.70 fold, p < 0.001, t-test) and Thailand (2.25 fold, p < 0.001, t-test) is also consistent with higher *P. vivax* transmission and a more rapid acquisition of antibodies in this population.

Antibody levels to 34 Pv antigens negatively correlated to the time since previous infection (p < 0.001, Kruskal-Wallis test, S7 Table) and positively correlated to the number of blood-stage detections during follow up in all three cohorts (p < 0.001, Kruskal-Wallis test, S8 Table). We selected three antigens (PVX_094255B (PvRBP2b), PVX_090240 (PvCyRPA) and PVX_099980 (PvMPS1$_{19}$)) as examples due to their documented immunogenicity [23,24,27]. Fig 3 shows the association between the antibody response measured at the final time point and (i) time since previous infection; (ii) number of qPCR positive blood-stage infections detected; and (iii) age. Similar patterns are seen across antibodies to all antigens in all regions (S4–S6 Figs). Antibody responses decrease with time since previous infection; increase with the number of blood-stage infections detected during one year of follow-up; and increase with age. Anti-CyRPA antibody responses in Brazil provide a notable exception, not exhibiting any significant association with the studied factors.

## Marker selection to detect exposure in the previous 9 months

Antibody responses were assessed by their classification performance for detecting blood-stage *P. vivax* infection within the previous 9 months for each study site (with the results for Thailand and Brazil previously reported [11]). The sensitivity and specificity trade-off of each antigen was tested through the area under the ROC curve (AUC). The antibody response to

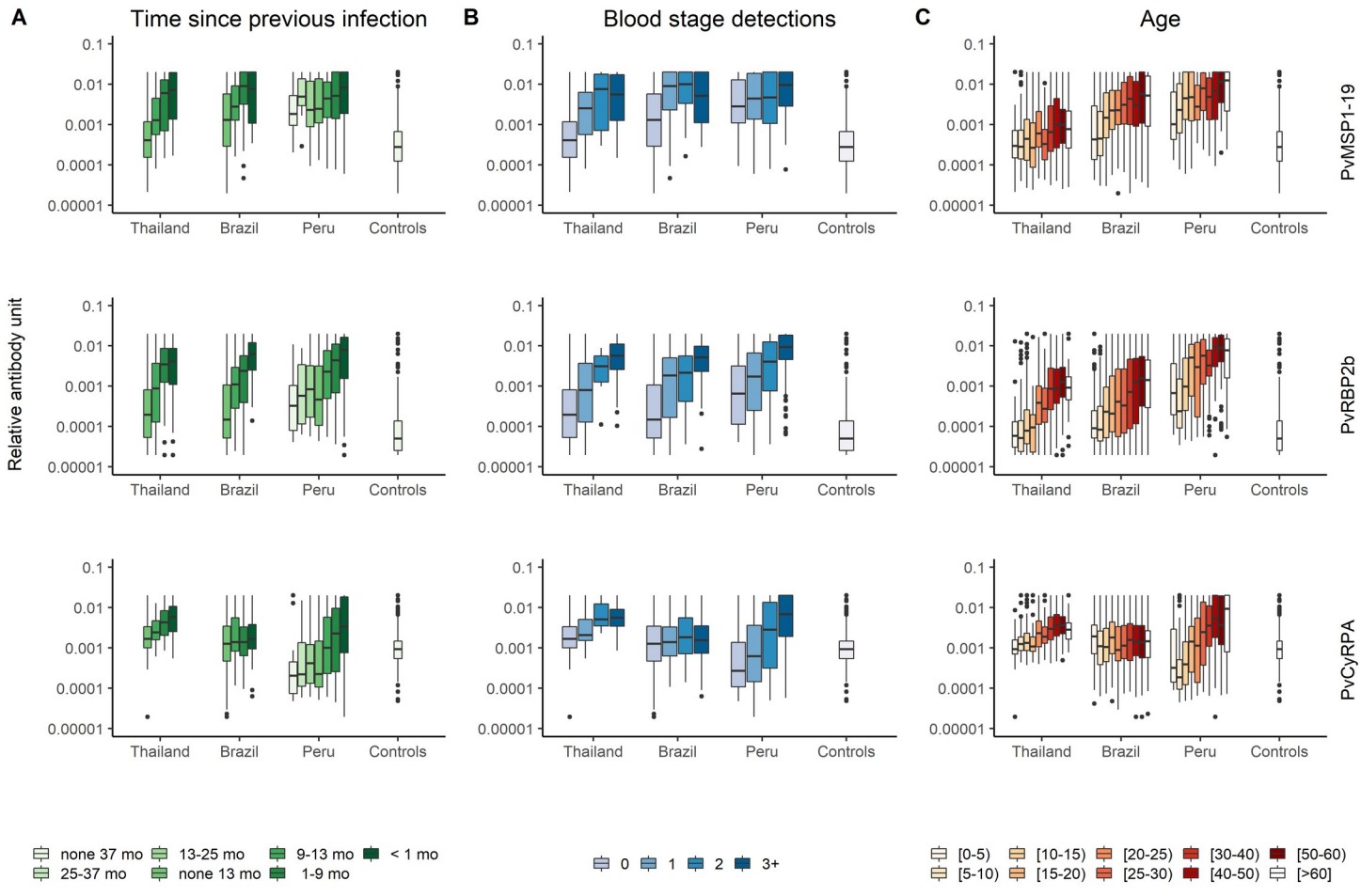

**Fig 3. Association of antibody responses to PvMSP1₁₉, PvRBP2b and PvCyRPA with epidemiological features.** (A) Antibody titers and time since previous infection during follow up. (B) Antibody titers and number of blood stage infections detected by qPCR. (C) Antibody titers across age groups. Note the controls are grouped together regardless of age. Mo: number of months since previous infection.

PVX_094255 (PvRBP2b₁₆₁₋₁₄₅₄ and PvRBP2b₁₉₈₆₋₂₆₅₃) was the top marker for the three study sites and for the combined data set (S2 and S9 Tables). Across all populations, the top ranking markers were the antibody responses to PVX_094255 (PvRBP2b), PVX_121920 (PvRBP2a), PVX_087885 (PvRAMA), PVX_099980 (PvMSP1₁₉) and PVX_097715 (hypothetical PvMSP3) (S2 Table). Although the cohort studies were carried out in co-endemic sites for *P. vivax* and *P. falciparum*, we did not find an association between the antibody levels to the top five proteins and recent *P. falciparum* infection (S7 Fig). However, the number of individuals with at least one *P. falciparum* infection in the preceding 13 months of last time point was low: Thailand n = 9, Brazil n = 17, Peru n = 61.

The antibody levels to the two constructs of PvRBP2b were highly correlated (Pearson's r = 0.83, p < 0.001), so we excluded the PvRBP2b construct with the lower level of accuracy (PvRBP2b₁₉₈₆₋₂₆₅₃). At a population level, the performance of antibody responses to PVX_094255 (PvRBP2b₁₆₁₋₁₄₅₄) for classifying recent infections was better in Thailand (AUC = 0.83) and Brazil (AUC = 0.79), than in Peru (AUC = 0.68) (Fig 4A). Moreover, there were markers such as PVX_099980 (PvMPS1₁₉) that showed good performance only in

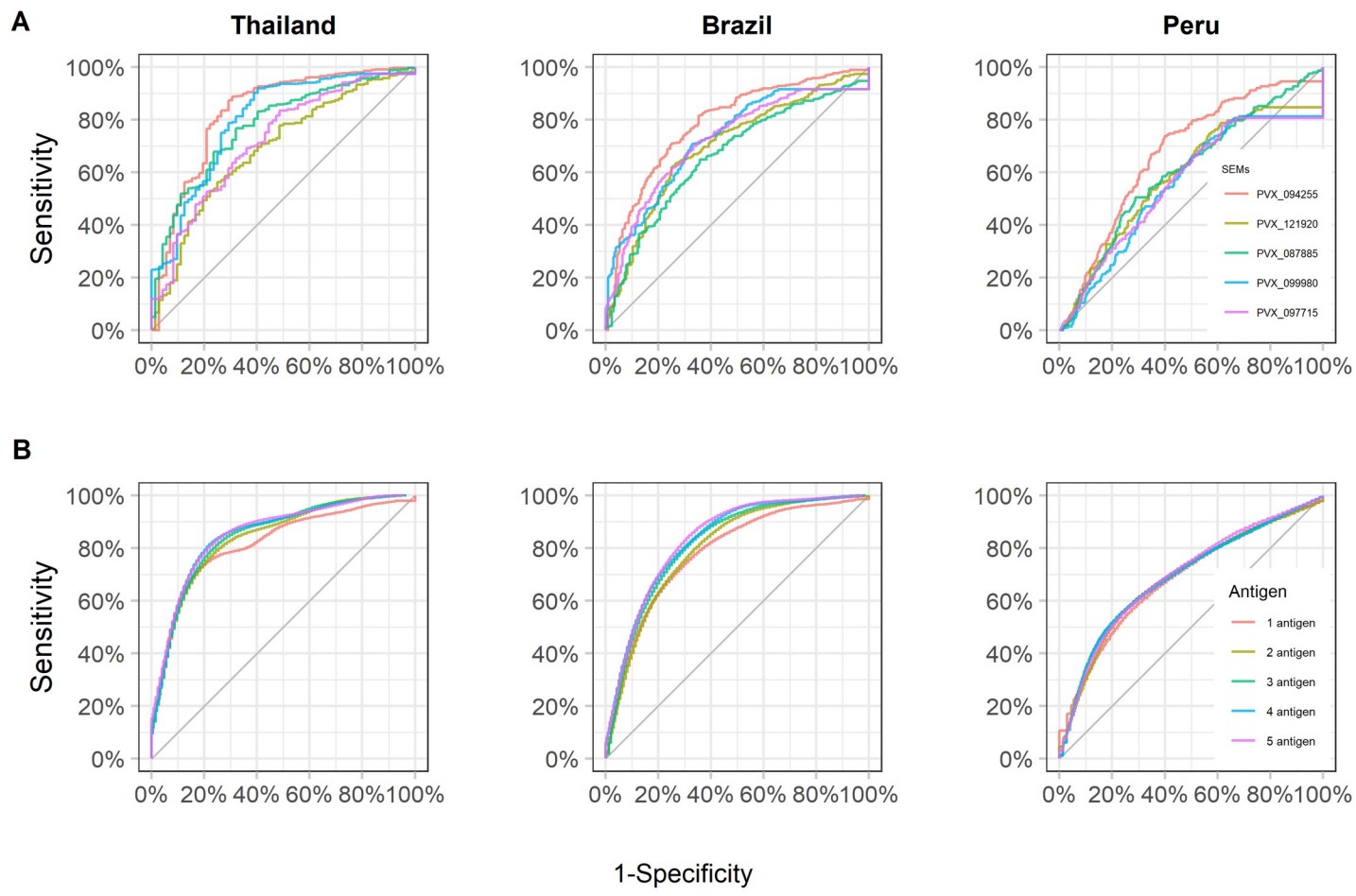

**Fig 4. Diagnostic performance for classifying recent exposure using antibody responses.** (A) Receiver operating characteristic (ROC) curve displays the diagnostic performance of 5 top antibody responses individually: PVX_094255 (PvRBP2b), PVX_121920 (PvRBP2a), PVX_087885 (PvRAMA), PVX_099980 (PvMSP1$_{19}$) and PVX_097715 (hypothetical PvMSP3) in the study sites. (B) ROC curves displaying the diagnostic performance to detect recent infections given by combinations of the top 5 antibody responses.

Thailand and Brazil, and others such as PVX_090240 (PvCyRPA) that performed well only in Peru. Evaluation of antibody responses to PVX_094255B (PvRBP2b), PVX_090240 (PvCyRPA) and PVX_099980 (PvMPS1$_{19}$), indicates that they can be used to detect previous *P. vivax* infections within a wide time range of 1–12 months, and not just the previously investigated 9 month target (S8 Fig) [11].

## Performance to detect recent exposure using combinations of antibody responses

While antibody responses to single antigens are informative for detecting recent exposure, a combination of antibody responses may lead to improved sensitivity and specificity in low-transmission regions [11]. Using a linear discriminant analysis (LDA) classification algorithm, models including up to five antibody responses were tested for maximizing information of exposure in the previous 9 months. The combinations of antibody responses that were most informative of recent exposure were cross-validated, and performance plotted using ROC curves. The top combinations of antibody responses per country were the following: Thailand: PvRBP2b, PvRBP2a, Pvs16, histone-lysine N-methyltransferase SET10 and PvMSP1$_{19}$

(AUC = 0.87); Brazil: PvRBP2b, hypothetical PvMSP3, PvRAMA, PVX_091710 and PVX_094830 (AUC = 0.85); Peru: PvCyRPA, PvRBP1a, PVX_090970, and the members from the *P. vivax* Tryptophan Rich Antigens (PvTRAG or Pv-fam-a) family PVX_096995 and PVX_092995 (AUC = 0.72). The top combinations of antibody responses for the combined data was PvRBP2b, Pvs16, PVX_091710, PVX_084720 (PTEX150) and PVX_094830 (AUC = 0.86). As expected, the multi-antibody response models performed better than the single antibody ones; although to a less extent for Peru (Fig 4B). We further analysed the top combinations of antibody responses for each study site stratified by two age categories: 0–15 and 15+ years old (S9 Fig). In the cohorts of Thailand and Brazil, the diagnostic performance was slightly higher in the group of 0–15 years old (Thailand $_{0-15}$ AUC: 0.90, Brazil $_{0-15}$ AUC: 0.87) than the group of 15+ years old (Thailand $_{15+}$ AUC: 0.83; Brazil $_{15+}$ AUC: 0.81). In both Cahuide and Lupuna, people older than 15 years old (Cahuide: $_{15+}$ AUC: 0.71, Lupuna: $_{15+}$ AUC: 0.66) were better classified than younger people (Cahuide: $_{0-15}$ AUC: 0.51; Lupuna: $_{0-15}$ AUC: 0.62).

## Determinants of variation of antibody responses

When focusing on PVX_094255B (PvRBP2b), PVX_090240 (PvCyRPA) and PVX_099980 (PvMPS1$_{19}$), the proportion of explained variance of antibody responses was different among sites and markers. For antibody responses to PvRBP2b the proportion of the variance explained was 37% in Thailand, 29% in Peru, and 21% in Brazil (Fig 5). Age (a proxy for lifetime exposure) accounted for the largest proportion of explained variance: 19% in Thailand, 9% in Brazil, and 16% in Peru. The time since previous infection explained 7% of the variance in Thailand, 7% of the variance in Brazil, and only 3% in Peru. The intensity of exposure in the previous year accounted for 12% of the explained variance in Thailand, 5% in Brazil, and 10% in Peru. The studied factors explained substantially less of the variance in anti-PvMSP1$_{19}$ antibody responses. This is most likely due to the high immunogenicity of PvMSP1$_{19}$ causing the generation of strong antibody responses after few infections. The explained variance of the antibody response to PvCyRPA represented 35% of the total variance in Peru, 22% in Thailand

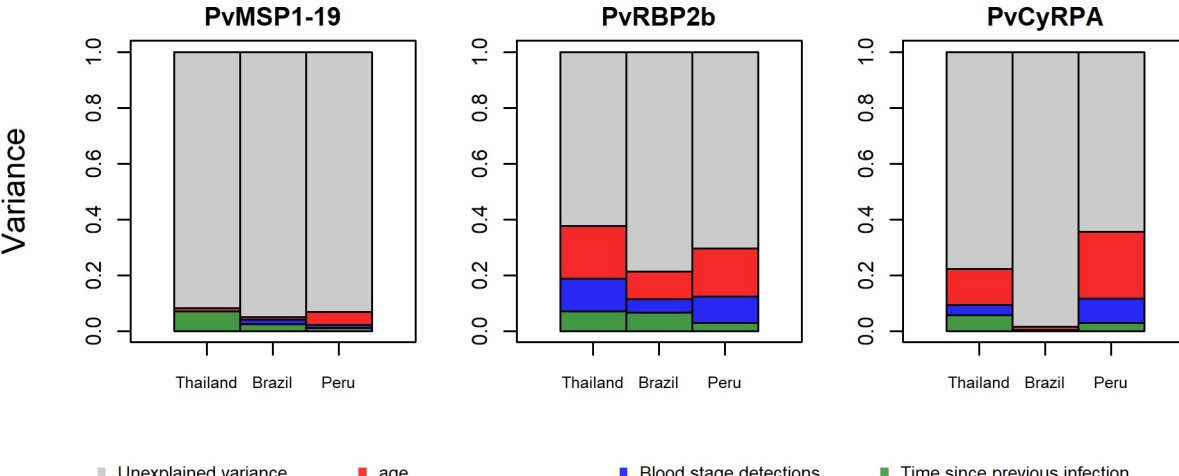

**Fig 5. Contribution of recent and lifetime exposure to the variance of antibody responses to PvMSP1$_{19}$, PvRBP2b and PvCyRPA across the study sites.** Panels represent the variance estimated for each marker. The area of each colored bar is proportional to the variance explained by known and unknown factors. Red bars: variance explained by individual's age. Blue bars: variance explained by number of blood stage detections by qPCR during 13 months of follow up. Green bars: variance explained by time since previous infection (days). Grey bars: unexplained variance.

and only 1.5% in Brazil. Age was the most important driver of the antibody response to PvCyRPA in Peru (0.35) followed by Thailand (0.13), and Brazil (0.01).

We assessed the suitability of serological markers for identifying recent infection across the three transmission settings (differentiated by cumulative PCR prevalence), by evaluating the geometric mean antibody levels to $PvMSP1_{19}$, PvRBP2b and PvCyRPA, and their performance at classifying recent exposure (Fig 6). The geometric mean antibody titer to $PvMSP1_{19}$ and PvRBP2b increased with the cumulative PCR prevalence, but did not for PvCyRPA. The performance to detect recent exposure (AUC value) of $PvMSP1_{19}$ and PvRBP2b was inversely correlated to cumulative PCR prevalence, but not for PvCyRPA. The variance explained by the time since previous infection decreased with the cumulative PCR prevalence.

With each cohort contributing a single data point, it is challenging to assess the significance of these relationships. Nonetheless, these exploratory findings lead us to think that these *P. vivax* SEMs are most suitable in low transmission settings where population level antibody titers are low, and a high proportion of the total variation in measured antibody titer is explained by the time since last *P. vivax* infection.

## Discussion

The application of serology to malaria control and elimination programs has been hampered by the lack of standardized techniques, well-developed antigenic markers, and validation in various transmission scenarios. With the potential for malaria elimination in many endemic countries, a tool able to both identify pockets of residual transmission and certify the malaria free status (in elimination settings) is urgently needed. Here, we evaluated the antibody responses to 34 novel *Plasmodium vivax* antigens in participants from a longitudinal cohort implemented in a high transmission setting in Peru, and made comparisons to data from low to moderate transmission settings in Thailand and Brazil [11].

In the three cohorts, individuals with blood-stage infections detected in the previous 9 months had significantly higher antibody titers than those with older infections; however, the performance of SEM to classify recent exposure was lower in Peru than in Thailand and in Brazil. The effect of malaria transmission on antibody responses is still poorly understood [12], our data showed that antibody responses to serological markers of recent exposure are influenced by the transmission intensity of the studied population. Whilst the geometric mean titer of most antibody responses was positively correlated with the cumulative PCR prevalence of each study site, the differences of antibody titers between positive PCR and negative PCR groups were highly significant in Thailand and Brazil but less pronounced in Peru, suggesting a slow decay of antibody responses in currently non-infected people in high transmission settings. This is also evidenced by the comparatively high antibody responses in Peruvians with an absence of blood-stage infections during one year of follow-up. Furthermore, our results supported the fact that the acquisition rate of antibody responses is faster in high transmission intensity sites, a phenomenon previously described. For example, King *et al.* showed that the antibody response repertoire to *P. vivax* and *P. falciparum* antigens was broader with corresponding higher antibody levels in subjects residing in high vs low transmission conditions [28]. Anti-malaria antibodies are mostly short-lived [29], but their longevity is antigen specific [30] and may vary according to the background immunity and transmission intensity of the studied population [31,32]. In the cases of *P. vivax* Merozoite Surface Protein-1 (PvMSP1), antibody responses can last up-to 30 years in sites where there has been sufficient levels of past exposure [33]. The antibody longevity is maintained by memory B cells that upon re-exposure rapidly proliferate and differentiate into antibody secretory cells, greatly boosting antibody levels [34]. In settings of low transmission, *P. vivax* infections appear to induce long-lasting B cell

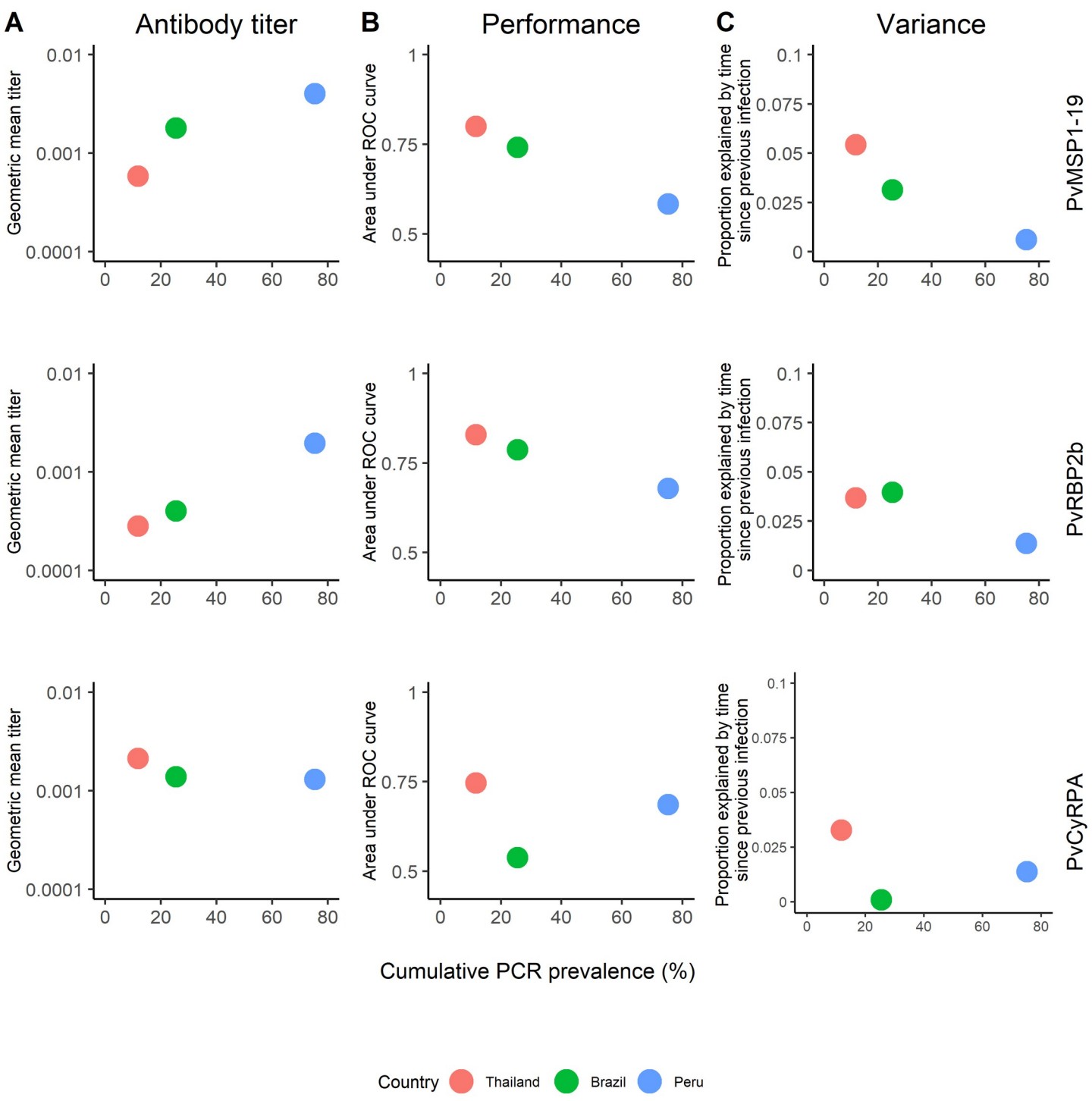

**Fig 6. Association of cumulative PCR prevalence and antibody responses to PvMSP1$_{19}$, PvRBP2b and PvCyRPA.** (A) Association of cumulative PCR prevalence and the geometric mean titer of antibody responses. (B) Association of cumulative PCR prevalence and the area under the ROC curve. (C) Association of cumulative PCR prevalence and the variance explained by the time since previous infection.

memory but with corresponding relatively short-lived antibody responses [9,10], although long-lived antibody responses have also been reported [35]. Although it has not been rigorously investigated, in high transmission settings both memory and antibody responses appear to be long-lived [36]. A plausible explanation of the high antibody levels in the Peruvian cohort is that frequent new infections and relapses would boost the antibody response baseline in the Peruvian population and that would result in a slow decay of these responses [37]. In light of these broad associations between transmission intensity and natural acquisition of immunity, we thus hypothesize that the differential performance of SEMs in the three study sites are likely due to the differences in malaria transmission intensity. In Thailand and Brazil, where the prevalence is low, it is likely that the immune repertoire is smaller and antibody responses are more short-lived. In Peru, the high antibody titers in the cohort individuals suggests a consistently high exposure to *P. vivax*, which is expected to elicit a broad repertoire of long-lived antibodies.

In our study, antibody titers increased with individuals' age. There are two possible explanations. Antibody levels may increase with age reflecting the cumulative nature of the immune response to *P. vivax*, and highlighting age as a surrogate marker of life-time exposure. This relationship has been exploited to assess malaria transmission intensity in endemic settings through modelling seroconversion rates, which are closely correlated with the entomological inoculation rate [12,38]. On the other hand, the association of antibody responses and age may also reflect the epidemiological characteristics of each population. The transmission of *P. vivax* in Thailand is mostly related to forest jobs in adults [35], whereas in Brazil and Peru, the transmission is both peri-domestic and work-related [18,39]. Our linear regression model showed that the time since previous infection and the intensity of exposure in the last year were important drivers of the antibody response to the top markers, PvRBP2b and PvMSP1$_{19}$, in Thailand and Brazil, whereas the cumulative exposure, as indicated by individuals' age, explained most of the variance in antibody responses in Peru. In a previous study on the baseline of the Peruvian cohort, Rosas-Aguirre *et al.* reported the association of antibody responses to PvMSP10 with *P. vivax* infections in the previous 6 months and individual's age [40]. PvMSP10 was not included in our panel of SEMs; however, the four PvMSP markers that ranked within the top 8 SEMs for classifying recent exposure in Peru (PvMSP3.3; PvMSP3.10; and two PvMSP7 markers) also showed association to individuals' age. Our results confirm the previous finding of a strong influence of age on PvMSP antigens' performance as a marker of recent exposure to *P. vivax*. Thus, the previous life-time exposure resulting from sustained transmission would influence SEMs performance in high-transmission settings.

Besides previous life-time exposure, the immunogenicity of markers and their interplay with transmission intensity were factors that affected the SEM performance. For example, the levels of antibody responses to PvRBP2b were positively correlated with the number of blood-stage detections and time since previous infection in the three cohorts, indicating that multiple antigen exposure is needed in order to boost an antibody response, and thus showcasing the moderate immunogenicity of this antigen. PvRBP2b belongs to PvRBP family of membrane proteins involved in the irreversible adhesion to reticulocytes [41]. Reports on naturally acquired immunity against PvRBP2b have shown that antibody responses correlated with age, infection status and clinical protection [24,27,42]. Previous reports have shown that antibodies responses against other members of the PvRBP family (i.e. PvRBP1a, PvRBP2a, PvRBP2c and PvRBP2-P2) also correlated with intensity of exposure and increased at a slower rate than antibodies against highly immunogenic antigens such as anti-PvDBPII [43] or PvCSP [23]. Recently, longitudinal studies have provided estimates of the longevity of antibody response against PvRBP2b, estimating a mean half-life of ~ 3.8 months in cohorts from Thailand and Brazil [11], indicating the moderate longevity of antibodies even in low transmission settings.

Importantly, Longley *et al.* showed that the antibody response to PvRBP2b had low reactivity in the malaria-naïve negative control group, which likely also has a strong impact on classification performance [11]. These characteristics make anti-PvRBP2b antibodies the best performing SEMs for classifying recent exposure in all three cohorts.

The use of highly immunogenic antigens such as PvMSP1$_{19}$ in serological tools has been advised in areas of low transmission, where the long-term persistence of antibody levels may be suitable for estimating changes in transmission intensity [12]. Lowly immunogenic markers with shorter half-life antibodies like PvCyRPA would be suitable for estimating the time of the last infection in high endemicity scenarios. Our antigen panel consists of constructs whose antibody responses longevity spanned up to 6 months in the absence of detectable recurrent infections in longitudinal cohorts in two low transmission sites [32] and whose performance has been validated to detect recent exposure in the context of low transmission intensity [11]. Our new findings suggest that the reactivity to these markers may not necessarily reflect recent exposure in high-transmission scenarios like Peru, and that the constant, long-term exposure to *P. vivax* may result in the long-lived IgG profiles in the absence of ongoing *P. vivax* infection in that cohort. Measurement of antibodies in longitudinal samples from the Peruvian cohort in the future would confirm our results and shed light on the antibody kinetics to *P. vivax* in a higher transmission scenario.

Although our work did not address the parasite genetic diversity on these cohorts, we reckon that parasite genetics could play a role in the variation of antibody responses between cohorts [44,45], e.g. antibody responses to PvCyRPA changed between countries. In the future, it would worth assessing the genetic variation of the proposed SEMs and their impact in diagnosis in different endemic settings. Our approach's advantage is that we use many antigens that produce redundancy, i.e. if one antigen does not work in a particular site, other antigens could still provide information. Another limitation of our work is the fixed cohort design in Peru. Migration affected the cohort's design with loss of follow up and entry of new residents (who were not included in the study). Migration may have affected the incidence and prevalence either by introducing new strains or by the introgression of malaria-naïve individuals [19]. Unlike Lupuna, Cahuide is vulnerable to imported malaria due to work related-human mobility and social interactions within and between other endemic communities [18]; however, the transmission dynamics in the Peruvian amazon goes beyond our work aims. There are a number of potential applications for SEMs in malaria control and elimination programs. At a population level, SEMs can be useful to identify pockets of residual transmission in geographic regions with heterogeneous or focalized transmission and measure the impact of interventions [46–48]. These use cases can be applied in contexts of medium to very low transmission levels. At an individual level, SEMs could be implemented in serological test and treat (PvSeroTAT) strategies for preventative treatment of *P. vivax* with Primaquine or Tafenoquine in elimination campaigns [11]. Such an intervention could reduce *P. vivax* cases with efficacy similar to mass drug administration (MDA), but with the benefit of lower rates of overtreatment.

Finally, our study highlights the importance of an in-depth immunological dissection of the antibody signatures in diverse transmission contexts of *P. vivax*. Such an exploration will allow us to understand the antibody kinetics in a context of ongoing transmission and to select suitable markers of recent exposure in high transmission settings. Field studies have shown that IgG subclass profiles to *P. falciparum* [49] and *P. vivax* [50–52] differed among malaria antigens and that the subclass predominance is influenced by age and increasing exposure to infection. High IgG1 and IgG3 levels have been associated with long-term exposure to *P. vivax* [42,52]. To improve performance of SEMs, further information could potentially be gained by testing IgG subclass responses and their ability to discriminate recent from distant exposure to *P. vivax* in areas with history of high past transmission.

In summary, we have shown that the antibody responses to SEMs reflect exposure in the previous 9 months in areas of low transmission, whereas the responses will mirror a combination of both recent and cumulative exposure in areas of high transmission. By combining both highly and less immunogenic antigens, our panel is able to detect recent exposure in low transmission or pre-elimination settings, providing a suitable tool for population-level surveillance.

## Supporting information

**S1 Fig. Study design for the retrospective analysis of 13 months of follow-up.** Five hundred ninety (590) individuals from the Peruvian cohort were included from which 350 were followed up to 37 months, and 240 individuals were followed for the last 13 months of the study. For the Brazilian (13 months) and Thai (14 months) cohort, 925 and 826 individuals were included, respectively. In the three cohorts, a monthly blood sample was taken for diagnosis by light microscopy and qPCR. "0" denotes the last time point of following up where a serum sample was taken for Luminex assay.
(TIF)

**S2 Fig. Distribution of antibody responses to 34 Serological markers of exposure in the Peruvian communities of Cahuide (CAH) and Lupuna (LUP).** Red dot indicates geometric mean titer (GMT) in each group. *** = significant difference between GMT of CAH and LUP, $p < 0.001$, t-test.
(TIF)

**S3 Fig. Pearson correlation between antibody responses to 34 Serological markers of exposure in the Peruvian cohort.**
(TIF)

**S4 Fig. Association between antibody responses to 34 *P. vivax* antigens and time since previous *P. vivax* blood-stage infection.** Mo: number of months since previous infection.
(TIF)

**S5 Fig. Association between antibody responses to 34 *P. vivax* antigens and number of blood-stage infections detected by qPCR.**
(TIF)

**S6 Fig. Association between antibody responses to 34 *P. vivax* antigens and age.** Note the controls are grouped together regardless of age.
(TIF)

**S7 Fig. Association between antibody responses to top five *P. vivax* antigens and time since last *P. falciparum* blood-stage infections.** Antibodies responses to PvRBP2b (PVX_094255), PvRBP2 (PVX_121920), PvRAMA (PVX_087885), PvMSP1$_{19}$ (PVX_099980) and PvMSP3 (PVX_097715, hypothetical PvMSP3). Note that all individuals with at least one *P. vivax* infection were removed from the analysis. There were not significant differences between individuals with *P. falciparum* PCR detection in the last 9 months and people with no *P. falciparum* infection.
(TIF)

**S8 Fig. Association of diagnostic performance of antibody responses in different timeframes with the distribution of time since the previous infection.** Area under the ROC curve (AUC) values were calculated using data of single antibody responses to PVX_094255 (PvRBP2b), PVX_090240 (PvCyRPA) and PVX_099980 (PvMPS1$_{19}$) to detect infections in each timeframe. Grey bars indicate the frequency of time since previous *P. vivax* infection in

each study site.
(TIF)

**S9 Fig. Diagnostic performance for classifying recent exposure using antibody responses stratifying by age groups.** ROC curves displaying the diagnostic performance to detect recent infections given by combinations of the top 5 antibody responses in two age groups: 0–15 years old and >15 years old. Peru cohort data was analyzed according to the Cahuide and Lupuna communities. The turquoise curve represents the ROC curve in individuals older than 15 years old. The red curve represents the ROC curve in individuals younger than 15 years old. Thailand: $_{0-15}$ AUC: 0.90, $_{15+}$ AUC: 0.83; Brazil $_{0-15}$ AUC: 0.87, $_{15+}$ AUC: 0.81; Cahuide: $_{0-15}$ AUC: 0.51, $_{15+}$ AUC: 0.71; Lupuna: $_{0-15}$ AUC: 0.62, $_{15+}$ AUC: 0.66.
(TIF)

**S1 Table. Epidemiologic characteristics of the study sites and participants.**
(DOCX)

**S2 Table. Characteristic of evaluated constructs.**
(DOCX)

**S3 Table. Association of antibody responses with current *P. vivax* infection in the Peruvian cohort.**
(DOCX)

**S4 Table. Correlation between antibody titers and age.**
(DOCX)

**S5 Table. Multivariate linear regression model explaining the antibody levels in the Peruvian cohort.**
(DOCX)

**S6 Table. Geometric mean titer of 34 SEM across the study sites.**
(DOCX)

**S7 Table. Overall antibody response and time since previous infection.**
(DOCX)

**S8 Table. Overall antibody response and number of detected blood- stage infections.**
(DOCX)

**S9 Table. Top antibody responses for classifying recent infections.**
(DOCX)

## Acknowledgments

We acknowledge the field teams that contributed to sample collection and qPCR assays, including Carlos Fernandez-Miñope, Katherine Alcedo, Jhonatan Alarcon-Baldeon, Juan Jose Contreras-Mancilla. We thank Wai-Hong Tham, Eizo Takashima, Takafumi Tsuboi, Matthias Harbers, Chetan Chitnis and Julie Healer for kindly provide the proteins used in this study. We acknowledge Connie Li-Wai-Suen for writing the R script to convert MFI to RAU. We thank Christopher King (Case Western Reserve University) for provision of the PNG control plasma pool.

## Author Contributions

**Conceptualization:** Michael T. White, Rhea J. Longley, Alejandro Llanos-Cuentas, Dionicia Gamboa, Ivo Mueller.

**Data curation:** Jason Rosado.

**Formal analysis:** Jason Rosado, Michael T. White, Rhea J. Longley.

**Funding acquisition:** Marcus Lacerda, Wuelton Monteiro, Joseph M. Vinetz, Ivo Mueller.

**Investigation:** Jason Rosado, Michael T. White, Jessica Brewster, Alejandro Llanos-Cuentas, Joseph M. Vinetz, Dionicia Gamboa, Ivo Mueller.

**Project administration:** Marcus Lacerda, Wuelton Monteiro, Jetsumon Sattabongkot, Mitchel Guzman-Guzman, Alejandro Llanos-Cuentas, Joseph M. Vinetz, Dionicia Gamboa.

**Supervision:** Michael T. White, Rhea J. Longley, Dionicia Gamboa, Ivo Mueller.

**Visualization:** Jason Rosado, Michael T. White, Ivo Mueller.

**Writing – original draft:** Jason Rosado.

**Writing – review & editing:** Jason Rosado, Michael T. White, Rhea J. Longley, Marcus Lacerda, Alejandro Llanos-Cuentas, Joseph M. Vinetz, Dionicia Gamboa, Ivo Mueller.

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
