## [Decision Letter · Decision Letter 0]

19 Oct 2020

Dear Mr Rosado,

Thank you very much for submitting your manuscript "Heterogeneity in response to serological exposure markers of recent Plasmodium vivax infections in contrasting epidemiological contexts" for consideration at PLOS Neglected Tropical Diseases. As with all papers reviewed by the journal, your manuscript was reviewed by members of the editorial board and by two independent reviewers. In light of the reviews (below this email), we would like to invite the resubmission of a revised version that takes into account the reviewers' comments. 

We cannot make any decision about publication until we have seen the revised manuscript and your response to the reviewers' comments. Your revised manuscript is also likely to be sent to reviewers for further evaluation.

Sincerely,

Margaret A Phillips, Ph.D.

Deputy Editor

Margaret Phillips

Deputy Editor

Reviewer's Responses to Questions

**Key Review Criteria Required for Acceptance?**

**Methods**

-Are the objectives of the study clearly articulated with a clear testable hypothesis stated?

-Is the study design appropriate to address the stated objectives?

-Is the population clearly described and appropriate for the hypothesis being tested?

-Is the sample size sufficient to ensure adequate power to address the hypothesis being tested?

-Were correct statistical analysis used to support conclusions?

-Are there concerns about ethical or regulatory requirements being met?

Reviewer #1: Minor comments

- Line 87- what about RDT? As you are advocating a serological rapid test, would be good to compare it to the equivalent rapid test that is currently available

- Line 135- how were the volunteers chosen? Randomly? What was the loss to follow up- was that the reason for the extra participants in the third year?

- When stating the numbers of active case detection visits- presumably this was maximum? Not everyone would have been available for all. 

- State a bit more information re. the Thai and Brazilian cohorts- all ages, randomly selected? Why weren’t the full cohorts from Longley et al included?

- Could you provide a justification for the use of the linear model? As it is a cohort I would expect to see a model accounting for time under observation- but I think you have reduced your explanatory variables to binary? At some point you report odds ratios suggesting you have used a cut off (not defined) to designate responses as negative/positive- would this be a more appropriate way to look at risk factors (though I acknowledge you lose information by reducing responses to binary)? I also feel that including age as a continuous variable in the model may be inappropriate- I think a categorical variable would more appropriately capture the dynamic changes over age group. 

- Line 184- it is not clear to me what this analysis entails- by taking into account, do you mean adding in to the model?

- Line 185: D0 is a bit confusing as it’s actually at the end of the cohort- consider renaming

- Line 189- couldn’t Peru have been longer because of the three year cohort? It does seem further data is used in your figures?

- Line 188: Is there a reason 9 months was picked as the cut off for this?

- Could you give a fuller explanation of disjoint training (line 196)?

- Is this a cut off determined from your negative samples or from the models? How was the cut off value determined?

- Line 199- clarify whether age was used as a linear var- it seems in the supplementary that it was also log?

Reviewer #2: Throughout the objectives, objectives are clear, well-defined, and analyzed appropriately. 

All populations are well-described in text, or in refs (prior cohorts), and all ethical issues are covered.

** Edits required:

Line 134. The description of both "home-to-home" and "community-based" screening should expanded, to make the sampling frames and enrollment processes clear.

Line 135. While recent anti-malaria drug use was an exclusion criterion, what the standard-of-care during this cohort, and what exclusions if any existed for antimalarials during the follow-up period? (Table 1).

Line 187. Was sensitivity analysis performed for the choice of 9-months as the cutpoint for "recent exposure" ?

Line 195. References should be added for LDA (in general), and the justification for the 1/3+2/3 split and 200 cross-validations.

A overview map of the study region would be welcomed, but is not absolutely essential.

** Minor edits: 

Line 128: "Malaria" need not be capitalized.

**Results**

-Does the analysis presented match the analysis plan?

-Are the results clearly and completely presented?

-Are the figures (Tables, Images) of sufficient quality for clarity?

Reviewer #1: - Line 214: ratio, not rate

- Line 215: if short duration in a community is a risk factor- does this not suggest that they were infected elsewhere? This may limit the usefulness of including them in a population study looking at transmission in a given setting- could be something to bring up in the discussion re the importance of understanding the population context. 

- Line 224- given that these are samples repeated on individuals, who presumably were not treated (?), could you state how many individuals were infected at any point during the cohort, rather than just the number of positive samples

- Table 1: time since previous infection numbers are confusing- is this at the end of the follow up?

- Why are you presenting statistical differences between the two areas sampled- do you then include this in the model? Or is it to highlight a higher and lower transmission place? Maybe make this clearer in the text- you subsequently put all the Peru data together but maybe it could be kept separate considering the differences in transmission intensity between the two settings (maybe too few samples?)

- It seems odd to me that you’re presenting prevalence throughout rather than incidence- are you considering each monthly visit as a cross-sectional survey, rather than a cohort? Is there any instances in your analysis where you should be account for the fact that you have multiple infections from the same person? And also refer to previous comment re. whether you know if these are multiple infections or the same infection detected on subsequent visits. 

- Line 241- why do you pick the age of 40 here?

- When you are talking about infected in the previous x days- make clear that this is at the end of the cohort (if I have understood correctly)- again, D0 makes this a bit confusing- which you’d think would be at the beginning, not the end

- If you are using an odds ratio- then you must have a binary variable for your antibody responses- but you have not mentioned determining a cut off anywhere? I couldn’t find reference for this logistic model in your methods.

- Fig 2- your age brackets are misaligned

- Relative antibody units are all negative- is this expected?

- Line 251: are these results from models with just the one explanatory factor? I think age should be controlled for a priori as you are probably looking at previous exposure- which is potentially not what you are interested in here- do the SEMs get less specific with age group?

- In para staring line 251 you talk about previous 6 months- but previously it has been 9 months- why the discrepancy? Please justify the 9 month cut off. 

- Line 271: what does positively correlated to time since previous infection mean- that would suggest to me that the longer ago the infection, the higher the antibody responses- perhaps needs rewording?

- Line 272: what are blood stage detections? PCR +ve?

- Figure 6: I think your Y axis need to highlight that they’re looking at the association with those things- not actually those things- at first I was surprised that Peru seemed to have lower mean titres than the other settings. 

- Figure 6- these dots are very large- is that representative of any uncertainty? Or are they just large?

Reviewer #2: ** Edits required: 

Line 236. As the geo mean responses overlapped, the villages were combined for analysis; however, the residence time in the villages differs sharply. Was this covariate added in the multivariable models?

Line 244 (Figure 2). The metric "immunogenicity" (y-axis) on should be clarified- is it quantitative or just a general sorting?

Line 323ff. Are the 95% CIs for the "percent explained variation" estimates overlapping here? That is, are these differences meaningful?

Line 339ff. While the authors do state the limitations and caveats of these 3-data point comparisons, these comparisons should be further noted to be "exploratory." The relapse rates, EIRs, and recent transmission regimes are so radically different in these settings make comparisons difficult

**Conclusions**

-Are the conclusions supported by the data presented?

-Are the limitations of analysis clearly described?

-Do the authors discuss how these data can be helpful to advance our understanding of the topic under study?

-Is public health relevance addressed?

Reviewer #1: - Line 381: you state that both memory and antibody cells appear to be long-lived- do you have a reference for this? You are measuring antibody cells- not memory cells- so could it not be that a higher response is developed in the higher transmission setting and therefore it may be waning but still appears high after a longer period post-infection- this is described in the following sentences in your discussion. I’m not sure you know whether memory cells are responsible. 

- Line 401: I think this sentence needs rewording- if you didn’t test PvMSP10 then you can’t confirm those results- you could say that you see similar patterns with other MSP markers?

- Line 413- if antibodies have a mean half life of 4 months, won’t you potentially be missing these if you at people who had an infection 9 months ago? I’m still unclear on the justification of this time lapse

- Legend for time since previous infection- could you reverse your legend so that <1 month is at the other end to reflect the bar placement?

- Table 1: supplementary- are these logistic regressions controlled for any other variables?

- Table II: supplementary- Did you look at each of these risk factors separately? Or just put everything in the model at once?

- Table VI: It wasn’t clear to me why the negative controls were multiplied by 1000 and isn’t mentioned in methods

Reviewer #2: The conclusions largely supported, with one exception. 

** Edits required.

Discussion on the geographic variation in parasite populations should be expanded, as this has direct implications for assessing how much of the observed variation is due to transmission intensity as proposed, and how much may be due to immunogenicity of parasite sub-populations. This study cannot and was never intended to address this issue, but there should be greater consideration of this potential confounding.

https://www.ncbi.nlm.nih.gov/pmc/articles/PMC5552344/

https://www.ncbi.nlm.nih.gov/pmc/articles/PMC3713979/

https://www.ncbi.nlm.nih.gov/pmc/articles/PMC5347536/

https://www.ncbi.nlm.nih.gov/pmc/articles/PMC4021508/

https://www.ncbi.nlm.nih.gov/pmc/articles/PMC5809496/

**Editorial and Data Presentation Modifications?**

Reviewer #1: (No Response)

Reviewer #2: (No Response)

**Summary and General Comments**

Reviewer #1: This paper summarises a serological dataset from Peru and evaluates responses to a panel of P. vivax antigens- assessing correlation with age, number of infections and time since last infection. The paper aims to validate the use of specific antigens, or combinations of antigens, to be used as markers of recent exposure to P. vivax. The work mainly focusses on a cohort in Peru- but brings in previously analysed datasets from different transmission intensity settings in Thailand and Brazil. The paper is well written and interesting though there is a lot of information to summarise and this isn’t always achieved in the clearest way. I enjoyed the paper and think the data is important and the analysis justified. However, I have a few major comments which need addressing and quite a lot of minor comments which I think would make things clearer for the reader. 

Major comments

- The number of detectable infections doesn’t seem like a correct measure to me. As far as I understand it, the people in the cohorts were not treated as these infection measures were done by PCR (so likely with some time delay). In which case, what you are measuring might not be multiple infections, but the same chronic infection? It’s not mentioned whether each time they were positive was counted as a new infection or not- please clarify that in the methods. I would also consider removing this as a metric if it is that you cannot be sure whether each positive PCR is actually a new infection or not. Is there any data on treatment in the cohort? Did people oscillate between PCR negative and positive over different visits? It would be interesting to know more about who is most likely to be getting infections. 

- Given that the need for a tool to demonstrate malaria elimination is highlighted several times- I think the seemingly potential positive responses amongst your negative sera should be addressed. This obviously highlights the risk of non-specific responses which is not mentioned in the manuscript. In Table VI (supplementary): For some antigens, the negative controls have a higher mean titre than the samples from endemic countries- would you recommend the cessation of use of these antigens?

- The Brazil and Thai cohorts have been previously analysed in Longley et al. It feels like some of that analysis has been duplicated in this paper but without the Soloman Islands data- is there a reason that dataset has not been included (maybe the age range?)? I think at times the comparisons to the other cohorts feels a bit shoehorned in. 

- Age representing cumulative exposure vs risk behaviour- this is always a complicated thing to pull apart. From the paper, it seems that you don’t consider that age is necessarily a risk factor in Peru (you mention peri-domestic transmission)- in which case would it be useful to look at the ROC for different age categories in Peru. A seroTAT might not be useful looking in all age groups (if adults have been exposed multiple times and therefore have antibodies remaining from an infection of several years ago) but testing children in a household might be a more sens/spec way to identify households with ongoing transmission- alternatively the two settings in Peru might allow you to look at this. I recognise you have incorporated a large amount of variables into your models- but might it be more useful to stratify by age/location, rather than just include everything at the same time. It could be useful for operational guidance too- i.e. test children aged under XX- at the moment having stats for a doubling of age giving a 2.18 increase in antibody levels isn’t too useful (line 253).

Reviewer #2: Overall, this is a extremely well-structured and rigorous set of studies. The only limitations are related to clarity and overall length.

The manuscript overall is quite lengthy, and edits should be considered to maximize the impact.

PLOS authors have the option to publish the peer review history of their article (what does this mean?). If published, this will include your full peer review and any attached files.

Reviewer #1: No

Reviewer #2: No
---

## [Decision Letter · Decision Letter 1]

21 Jan 2021

Dear Mr Rosado,

We are pleased to inform you that your manuscript 'Heterogeneity in response to serological exposure markers of recent Plasmodium vivax infections in contrasting epidemiological contexts' has been provisionally accepted for publication in PLOS Neglected Tropical Diseases.

Best regards,

Margaret A Phillips, Ph.D.

Deputy Editor

Margaret Phillips

Deputy Editor

Reviewer's Responses to Questions

**Key Review Criteria Required for Acceptance?**

**Methods**

-Are the objectives of the study clearly articulated with a clear testable hypothesis stated?

-Is the study design appropriate to address the stated objectives?

-Is the population clearly described and appropriate for the hypothesis being tested?

-Is the sample size sufficient to ensure adequate power to address the hypothesis being tested?

-Were correct statistical analysis used to support conclusions?

-Are there concerns about ethical or regulatory requirements being met?

Reviewer #1: The authors have responded to previous suggestions made in the original review adequately

**Results**

-Does the analysis presented match the analysis plan?

-Are the results clearly and completely presented?

-Are the figures (Tables, Images) of sufficient quality for clarity?

Reviewer #1: The authors have responded to previous suggestions made in the original review adequately

**Conclusions**

-Are the conclusions supported by the data presented?

-Are the limitations of analysis clearly described?

-Do the authors discuss how these data can be helpful to advance our understanding of the topic under study?

-Is public health relevance addressed?

Reviewer #1: The authors have responded to previous suggestions made in the original review adequately

**Editorial and Data Presentation Modifications?**

Reviewer #1: I would still suggest the data points in figure 6 are unusually large..

**Summary and General Comments**

Reviewer #1: My original points have been addressed adequately. This is a large dataset which has been analysed robustly.

PLOS authors have the option to publish the peer review history of their article (what does this mean?). If published, this will include your full peer review and any attached files.

Reviewer #1: No

---

## [Editor Report · Acceptance letter]

10 Feb 2021

Dear Mr Rosado,

We are delighted to inform you that your manuscript, "Heterogeneity in response to serological exposure markers of recent Plasmodium vivax infections in contrasting epidemiological contexts," has been formally accepted for publication in PLOS Neglected Tropical Diseases.

Best regards,

Shaden Kamhawi

co-Editor-in-Chief

Paul Brindley

co-Editor-in-Chief
